# Hi-ArG: Exploring the Integration of Hierarchical Argumentation Graphs in Language Pretraining

**Jingcong Liang**[1], **Rong Ye**[1,3], **Meng Han**[2], **Qi Zhang**[1],
**Ruofei Lai**[2], **Xinyu Zhang**[2], **Zhao Cao**[2], **Xuanjing Huang**[1] **Zhongyu Wei**[1*]

[1]Fudan University, [2]Huawei Poisson Lab, [3]ByteDance
jcliang22@m.fudan.edu.cn,yerong@bytedance.com,
{qz,xjhuang,zywei}@fudan.edu.cn,
{hanmeng12,lairuofei,zhangxinyu35,caozhao1}@huawei.com

## Abstract

The knowledge graph is a structure to store and represent knowledge, and recent studies have discussed its capability to assist language models for various applications. Some variations of knowledge graphs aim to record arguments and their relations for computational argumentation tasks. However, many must simplify semantic types to fit specific schemas, thus losing flexibility and expression ability. In this paper, we propose the Hierarchical Argumentation Graph (Hi-ArG), a new structure to organize arguments. We also introduce two approaches to exploit Hi-ArG, including a text-graph multi-modal model GreaseArG and a new pre-training framework augmented with graph information. Experiments on two argumentation tasks have shown that after further pre-training and fine-tuning, GreaseArG supersedes same-scale language models on these tasks, while incorporating graph information during further pre-training can also improve the performance of vanilla language models. Code for this paper is available at https://github.com/ljcleo/Hi-ArG.

## 1 Introduction

Debating is a fundamental formal process to find solutions and gain consensus among groups of people with various opinions. It is widely accepted in politics (Park et al., 2015; Lippi and Torroni, 2016), education (Stab and Gurevych, 2014, 2017) and online discussions (Habernal and Gurevych, 2017). As more and more formal and informal debates are launched and recorded on the Internet, organizing arguments within them has become crucially valuable for automated debate preparation and even participation (Slonim et al., 2021).

Recently, language models have shown dominating advantages in various argumentation tasks (Gretz et al., 2020; Rodrigues and Branco, 2022). However, such tasks often rely on logical relations

---

*Corresponding author.

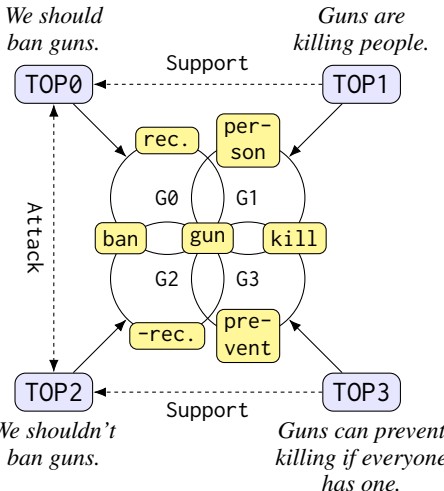

Figure 1: An example of Hi-ArG extracted from the conversation between Alice and Bob. TOP0 to TOP3 represent top nodes corresponding to different arguments. G0 to G3 represent intra-argument sub-graphs whose detailed structures are omitted. rec. and -rec. represent *recommend* and *not recommend* respectively.

between arguments and semantic relations between mentioned entities. To explicitly provide these relation links, Al-Khatib et al. (2020) borrowed the form of knowledge graphs and introduced the Argumentation Knowledge Graph (AKG) to represent entity relations extracted from arguments. They also applied AKG to fine-tune a language model for argument generation tasks (Al Khatib et al., 2021).

Despite the effectiveness of AKG, we argue that semantics within arguments can be better modeled. In this paper, we propose the Hierarchical Argumentation Graph (Hi-ArG), a graph structure to organize arguments. This new structure can retain more semantics within arguments at the lower (intra-argument) level and record relations between arguments at the upper (inter-argument) level. We claim that Hi-ArG provides more explicit information from arguments and debates, which benefits

language models in argumentation tasks.

For instance, consider the following conversation about whether guns should be banned:

- Alice: We should ban guns. Guns are killing people.

- Bob: No, guns can prevent killing if everyone has one, so we shouldn't ban them.

Alice and Bob have opposite opinions, yet their evidence focuses on the topic's same aspect (killing). In Hi-ArG (see Figure 1), all their claims and premises can live together and connect through inter-arg (supporting/attacking) and intra-arg (semantic relations between concepts like guns and killing) links.

To validate the power of Hi-ArG, we propose two approaches to exploit information from the graph. First, we introduce GreaseArG, a text-graph multi-modal model based on GreaseLM (Zhang et al., 2022) to process texts with sub-graphs from Hi-ArG simultaneously. Second, we design a new pre-training method assisted by Hi-ArG, which is available for both GreaseArG and vanilla language models. To examine the effectiveness of these approaches, we conduct experiments on two downstream argumentation tasks namely Key Point Matching (KPM) and Claim Extraction with Stance Classification (CESC).

In general, our contributions are:

- We propose Hi-ArG, a new graph structure to organize various arguments, where semantic and logical relations are recorded at intra- and inter-argument levels.

- We design two potential methods to use the structural information from Hi-ArG, including a multi-modal model and a new pre-training framework.

- We validate the above methods on two downstream tasks to prove that Hi-ArG can enhance language models on argumentation scenarios with further analysis.

## 2 Related Work

Computational argumentation is an active field in NLP focusing on argumentation-related tasks, such as argument mining (Stab and Gurevych, 2014; Persing and Ng, 2016; Habernal and Gurevych,

2017; Eger et al., 2017), stance detection (Augenstein et al., 2016; Kobbe et al., 2020), argumentation quality assessment (Wachsmuth et al., 2017; El Baff et al., 2020), argument generation (Wang and Ling, 2016; Schiller et al., 2021; Alshomary et al., 2022) and automated debating (Slonim et al., 2021). A majority of these tasks are boosted with language models recently (Gretz et al., 2020; Friedman et al., 2021; Rodrigues and Branco, 2022), and studies have discussed various ways to inject external argumentation structure or knowledge to such models (Bao et al., 2021; Dutta et al., 2022).

Meanwhile, several works have discussed proper methods to retrieve arguments from data sources (Levy et al., 2018; Stab et al., 2018; Ajjour et al., 2019), yet a majority of them keep the text form unchanged, resulting in loose connections between arguments. One of the methods to manage textual information is to represent it as knowledge graphs; however, popular knowledge graphs mainly record concept relations (Auer et al., 2007; Bollacker et al., 2008; Speer et al., 2017). To cope with this issue, Heindorf et al. (2020) proposed a knowledge graph that stores causal connections between concepts. Al-Khatib et al. (2020) further developed this idea and introduced the argumentation knowledge graph (AKG), which can be seen as the first approach to organize arguments structurally, and has been employed in a few downstream tasks (Al Khatib et al., 2021).

A flaw of the AKG is that relations between concepts are highly simplified, losing rich semantic information. The representation of semantic relations has been long studied as the task of semantic role labeling (SRL, Palmer et al. 2005; Bonial et al. 2014; Li et al. 2018; Shi and Lin 2019). While traditional SRL keeps the linear form of sentences, Banarescu et al. (2013) proposed abstract meaning representation (AMR), a graph-based structure to express semantic relations. Being similar to knowledge graphs, the idea becomes natural to record arguments in AMR graphs and organize them, then utilize them for downstream tasks in a similar way as typical knowledge graphs, especially accompanied with language models (Yang et al., 2021; Zhang et al., 2022; Amayuelas et al., 2022).

## 3 The Hi-ArG Framework

In this section, we introduce the detailed structure of Hi-ArG. We also propose a fully automated construction procedure applicable to different corpora.

## 3.1 Structure

As described in Section 1, Hi-ArG can be divided into the lower intra-argument level and the upper inter-argument level. In the remaining part of this paper, these two levels will be inferred as *intra-arg* and *inter-arg* graphs, respectively.

**Intra-arg Graph** This level represents the semantic relations within a particular argument, mainly related to semantic analyses like Abstract Meaning Representation (AMR, Banarescu et al. 2013). Since AMR is a highly abstract structure suitable for organizing semantic relations among concepts, it is introduced as the primary backbone of the lower level of the argumentation graph. More specifically, nodes in the intra-arg graph represent single semantic units connected with directed edges following the AMR schema. Arguments are linked to a node naming the top node, whose respective sub-graph fully represents the argument. More details of AMR concepts and their relation to intra-arg graph components can be found in Appendix A.

**Inter-arg Graph** The inter-arg level represents logical relations among arguments like supporting or attacking. Relation edges can connect arguments (top nodes) directly or through proxy nodes if multiple arguments need to be packed together. Formal logic and traditional argumentation theories (like the ones in Toulmin, 2003 or Budzynska and Reed, 2011) can be applied to this level.

The Hi-ArG structure has several advantages. First, note that different arguments can share one single top node if they have identical meanings. Such design allows efficient argument organization across paragraphs or documents. Second, if parts of sub-graphs of different top nodes are isomorphic, these parts can be merged, and thus the sub-graphs become connected. This way, arguments about common concepts may be grouped more densely, and debates around specific topics are likely to be associated.

## 3.2 Construction Pipeline

Based on the structure described above, We propose a brief construction pipeline for intra-arg graphs, as illustrated in Figure 2. Once the intra-arg graph is created, the inter-arg graph can be easily attached to the corresponding top nodes. Constructing inter-arg graphs usually depends on external annotations or argumentation mining results, and adding edges from such information is straightforward; hence, we will not cover the details here.

The construction pipeline can be split into three stages: (1) extracting, (2) parsing, and (3) merging.

**Extracting Stage** In the extracting stage, documents in the corpus are cut into sentences, and valid sentences are treated as arguments. The standard of valid arguments can be adjusted according to specific task requirements.

**Parsing Stage** As soon as arguments are extracted from the corpus, they are parsed into separate AMR graphs in the parsing stage. AMR parsing models can be applied to automate this stage. Besides parsing, additional operations can be performed during this stage, such as graph-text aligning (Pourdamghani et al., 2014).

**Merging Stage** After obtaining AMR graphs for each argument, the final merging stage will combine all these separate graphs into one Hi-ArG. This can be realized by iteratively merging isomorphic nodes or, more specifically, eliminating nodes with common attributes and neighbors.

## 3.3 Adapting Exploitation Scenarios

A few issues exist when applying Hi-ArG to more specific models and tasks. In this section, we introduce adaptations of the Hi-ArG structure to two common scenarios where the original one could cause problems. For instance, we apply these two adaptations to GreaseArG, one of our exploitation methods proposed in Section 4.

**Incorporating Text** Two issues arise when processing text information and its corresponding Hi-ArG sub-graph. On the one hand, this sub-graph may not be connected; on the other hand, multiple sentences in one text can point to the same top node, which could confuse each other. To resolve these problems, an extra link node is added for each sentence, along with a single edge connecting it to the corresponding top node. Furthermore, all link nodes are connected by another root node to ensure that the final graph is connected.

**Message Passing** A significant problem when processing Hi-ArG sub-graphs in message-passing models like graph neural networks (GNN) is that such graphs are directed, and directed edges cannot pass information in the reverse direction. To make message-passing algorithms effective, each edge is accompanied by a reversed edge, whose attribute is

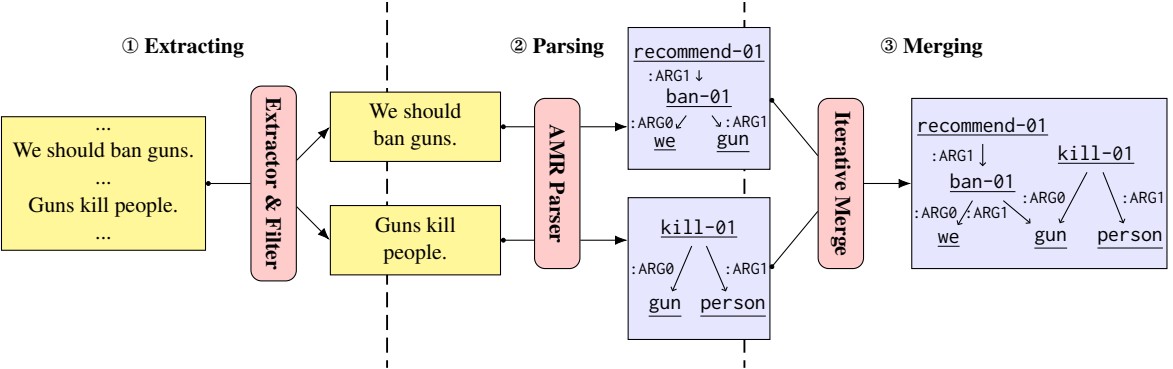

Figure 2: A construction pipeline for Hi-ArG intra-arg graphs.

also a reverse of the original edge. This is possible for intra-arg graphs because AMR can change the direction of edges by adding to or removing from the edge label the passive mark -of, and for inter-arg graphs, a similar approach can be used.

## 4 Exploiting Hi-ArG

Knowledge graphs can participate in various stages of standard NLP pipelines, such as information retrieval and model training. In this section, we introduce two methods of exploiting Hi-ArG on the model side. First, we present a model structure similar to GreaseLM (Zhang et al., 2022), referred to as GreaseArG, that simultaneously digests text and Hi-ArG data. We also propose further pre-training tasks specially designed for GreaseArG. Second, we develop a novel pre-training framework that modifies the method to construct training samples with another self-supervised pre-training task.

### 4.1 GreaseArG

Processing and understanding graph structures like argumentation graphs can challenge language models since they can only handle linear information like text segments. Following GreaseLM, GreaseArG uses graph neural network (GNN) layers as add-ons to facilitate this issue.

Figure 3 illustrates the structure of GreaseArG. Like GreaseLM, GreaseArG concatenates a GNN to transformer-based language models per layer and inserts a modality interaction layer between layers of LM/GNN. An interaction token from the text side and an interaction node from the graph side exchange information at each interaction layer.

Hi-ArG adaptations mentioned in Section 3.3 are applied so GreaseArG can properly process Hi-ArG sub-graphs. More specifically, for a series of sub-graphs, we create link nodes $S_1, S_2, \ldots$ con-

nected to the root nodes of each sub-graph, and an extra root node $R$ connected to all link nodes, as shown in Figure 3; furthermore, reversed edges (labels not displayed; with embeddings $e'_0, e'_1, \ldots$) are added to the connected directed graph to allow message passing in GNN.

Instead of using a specific prediction head as in GreaseLM, in GreaseArG, a cross-modal attention layer is appended after the output of the last LM/GNN layer, generating the final representation vectors of both text tokens and graph nodes. Another critical difference between vanilla language models and GreaseArG is that GreaseArG needs to generate word embeddings for special tokens in the AMR graph like :ARG0 and ban-01. This is achieved by increasing the vocabulary and adding slots for them.

In this paper, we adopt RoBERTa[1] (Liu et al., 2019) as the LM backbone of GreaseArG, and Graph Transformer (Dwivedi and Bresson, 2020) as the GNN backbone.

### 4.2 Pre-training Tasks for GreaseArG

Before heading towards the second exploitation method, we introduce more details about the pre-training tasks for GreaseArG. Since GreaseArG handles information from two modalities, we argue that pre-training this model should cover both. Here, we propose six tasks categorized into two classes: masking tasks and graph structure tasks. Each of the six tasks corresponds to a specific task loss; these losses are summed up to form the final pre-training loss. Figure 4 illustrates the two categories of tasks with inputs from Figure 3.

**Masking Tasks** Following Liu et al. (2019), we use Masked Language Modeling (**MLM**) as the

---

[1] Unless explicitly stated, RoBERTa will always refer to RoBERTa-base.

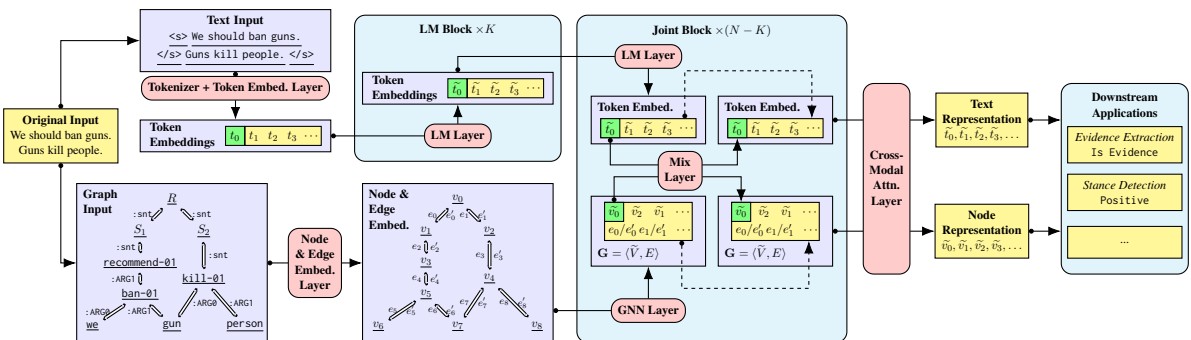

Figure 3: GreaseArG model structure when processing text with its Hi-ArG sub-graph. $\{t_0, t_1, \dots\}$ are token embeddings generated by the token embedding layer, whose transformed versions after at least one LM layer are denoted by $\{\tilde{t}_0, \tilde{t}_1, \dots\}$. $S_1$ and $S_2$ are link nodes added to connect subgraph roots `recommend-01` and `kill-01`, while $R$ is the extra root node added to connect link nodes $S_1$ and $S_2$; labels of reversed edges are not displayed for clarity. $\{v_1, v_2, v_3, \dots\}$ are node embeddings generated by the node embedding layer, whose transformed versions after at least one GNN layer are denoted by $\tilde{V} = \{\tilde{v}_1, \tilde{v}_2, \tilde{v}_3, \dots, \}$. $E = \{e_0, e'_0, e_1, e'_1, \dots\}$ are edge embeddings generated by the edge embedding layer (prime marks indicate reversed edges), which remain unchanged throughout the process. The embedded graph through one or more GNN layers is noted as $G = \langle \tilde{V}, E \rangle$ in the joint block.

text pre-training task. For AMR-based intra-arg graphs, Masked Components Modeling (MCM, Xia et al. 2022) can be treated as the graph version of MLM, where the model needs to predict attributes of masked components. In this paper, we consider masked node modeling (**MNM**) and masked edge modeling (**MEM**).

**Graph Structure Tasks** Graph structure tasks, focusing on learning structural information, consist of Graph Contrastive Learning (GCL), Top Order Prediction (TOP), and Edge Direction Prediction (DIR). **GCL** requires the model to discriminate nodes with randomly permuted attributes, following the modified GCL version from Zheng et al. (2022). **TOP** aims to predict the relative order of two consecutive link nodes from a document. **DIR** requires the model to determine the original direction of each edge in the bi-directed graph.

### 4.3 Augmenting Pre-Training with Relatives

Most language models today are pre-trained with chunks of text data, where continuous sentences from one or more documents are sampled and concatenated (Radford et al., 2018; Liu et al., 2019). When further pre-training GreaseArG, documents can refer to debates and other argumentation documents in the training corpus. Thus, sentences are sampled according to their original order continuously.

However, such a method does not exploit structural knowledge in Hi-ArG, a perfect tool for finding attacking or supporting statement pairs. Hence, we can augment original training samples with

topic-related sentences from Hi-ArG called *relatives*. In an augmented sample, each relative is linked to a specific sentence in one of the sampled documents and is inserted after that particular document. This provides another approach to generating pre-training samples.

For any specific sentence, its relatives are sampled from related sentences found in Hi-ArG, weighted by the two-hop similarity in the sentence-node graph $G_{\text{sn}}$. $G_{\text{sn}}$ is a bipartite graph between link nodes of sentences and nodes from the original Hi-ArG sub-graph, where each link node is connected to all nodes that form the semantic of its sentence; original edges that appeared in Hi-ArG are not included. The two-hop similarity between two sentences (link nodes) is the probability that a random walk starting from one link node ends at the other link node in two steps.

**Relative Stance Detection (RSD)** Suppose the topic and stance of sampled documents and relatives are known beforehand. In that case, we can classify each relative as supporting, attacking, or non-relevant concerning the document it relates to. Based on this observation, we propose a new pre-training task called Relative Stance Detection (RSD). In RSD, the model must predict the above stance relation between documents and their relatives. The final pre-training loss will include the corresponding loss of RSD, which is also computed by cross-entropy.

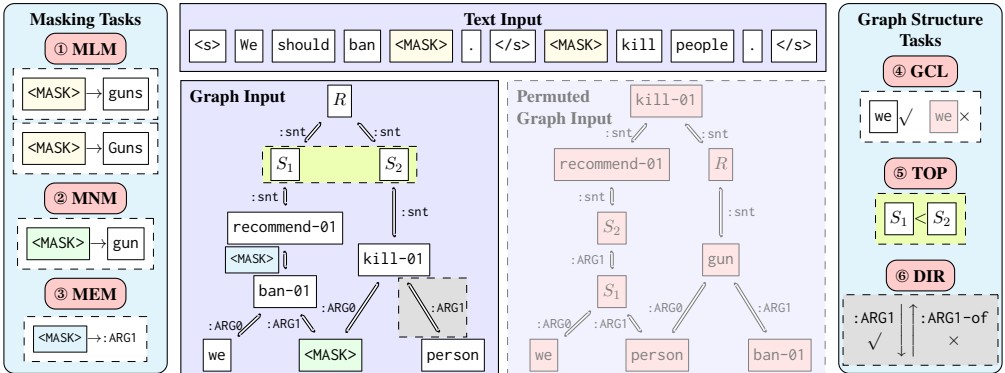

Figure 4: Pre-training Tasks for GreaseArG. **MLM**: Predict masked tokens in text input. **MNM**: Predict masked node attributes in graph input. **MEM**: Predict masked edge attributes in graph input. **GCL**: Predict whether a node belongs to the original or permuted graph input. **TOP**: Predict the relative order of consecutive link nodes. **DIR**: Predict the original edge direction. Note that the permuted graph input is only used by GCL.

## 5 Experiments

To examine our approaches to exploit Hi-ArG, we apply them to selected argumentation tasks and compare the results with previous work. All models are further pre-trained on argumentation corpus with generated Hi-ArG and then fine-tuned on specific downstream tasks. We report the main results at the end of this section.

### 5.1 Downstream Tasks

The performance of GreaseArG on argumentation tasks is evaluated on two downstream tasks — key point matching (KPM) and claim extraction with stance classification (CESC), covering argument pairing, extraction, and classification.

**Key Point Matching** This task (Bar-Haim et al., 2020a,b) aims to match long arguments with sets of shorter ones called key points. It is evaluated by the mean AP (mAP) of predicted scores between each candidate pair of arguments and key points. Some pairs are labeled as undecided if the relation is vague. Therefore, the mAP is further split into strict and relaxed mAP based on how to treat predictions on such pairs.

**Claim Extraction with Stance Classification** This (Cheng et al., 2022) is an integrated task that merges claim extraction and stance classification. In this task, the model needs to extract claims from a series of articles under a specific topic and then identify their stances. This task is evaluated as a 3-class classification task.

### 5.2 Dataset

In this section, we introduce the dataset used for further pre-training and task-related datasets for fine-tuning and evaluation. We construct each dataset's corresponding Hi-ArG using the graph-construction pipeline mentioned in Section 3.2. Appendix B describes the details of this procedure across datasets.

#### 5.2.1 Further Pre-training

Few works concern high-quality argumentation corpus sufficiently large for effective further pre-training. Among them, the *args.me* corpus (Ajjour et al., 2019) contains sufficient documents and arguments for further pre-training. Therefore, we construct the Hi-ArG of the *args.me* corpus to generate further pre-training samples.

Because KPM and CESC are highly related to inter-arg graphs, we only further pre-train models with intra-arg graphs to avoid bias towards these downstream tasks. Details about the construction of Hi-ArG for further pre-training can be found in Appendix C.

#### 5.2.2 Fine-tuning and Evaluation

We choose a specific dataset for each downstream task and construct its Hi-ArG, following the pipeline in Section 3.2. The training, dev, and test sets are processed separately, and we only construct intra-arg graphs like when building the further pre-training dataset.

**KPM: ArgKP-2021** The 2021 Key Point Analysis (KPA-2021) shared task (Friedman et al., 2021) provides a dataset for KPM called ArgKP-2021, covering 31 topics. Since they also report results

| Model | | KPM | | | CESC | |
|---|---|---|---|---|---|---|
| | | Strict mAP | Relaxed mAP | Mean of mAP | Macro $F_1$ | Micro $F_1$ |
| KPM | ModernTalk | 75.4 | **90.2** | **82.8** | – | – |
| Baseline | MatchTstm | 74.5 | **90.2** | 82.4 | – | – |
| CESC | Pipeline | – | – | – | 55.95 | 88.56 |
| Baseline | Multi-label | – | – | – | 60.25 | 91.22 |
| RoBERTa | no FP | $71.3_{\pm 2.8}$ | $87.7_{\pm 3.6}$ | $79.5_{\pm 3.2}$ | $59.01_{\pm 1.13}$ | $89.00_{\pm 0.50}$ |
| | FP w/o aug. | $72.6_{\pm 2.1}$ | $\underline{88.2}_{\pm 2.5}$ | $\underline{80.4}_{\pm 2.3}$ | $59.59_{\pm 1.29}$ | $89.58_{\pm 0.96}$ |
| | FP w/ aug. | $\underline{72.7}_{\pm 2.3}$ | $87.9_{\pm 1.6}$ | $80.3_{\pm 1.9}$ | $60.08_{\pm 0.95}$ | $89.36_{\pm 0.31}$ |
| | FP w/ mix | $72.2_{\pm 2.8}$ | $87.6_{\pm 1.5}$ | $79.9_{\pm 2.1}$ | $\underline{\mathbf{61.07}}_{\pm 0.80}$ | $\underline{\mathbf{90.28}}_{\pm 0.77}$ |
| GreaseArG | no FP | $72.5_{\pm 3.0}$ | $88.8_{\pm 2.1}$ | $80.6_{\pm 2.5}$ | $58.81_{\pm 0.82}$ | $88.75_{\pm 1.35}$ |
| | FP w/o aug. | $73.6_{\pm 1.3}$ | $89.3_{\pm 0.7}$ | $81.4_{\pm 0.6}$ | $59.03_{\pm 0.62}$ | $89.76_{\pm 0.34}$ |
| | FP w/ aug. | $\underline{\mathbf{75.8}}_{\pm 1.7}$ | $\underline{89.5}_{\pm 1.2}$ | $\underline{82.6}_{\pm 1.2}$ | $59.65_{\pm 1.27}$ | $89.61_{\pm 1.42}$ |
| | FP w/ mix | $71.5_{\pm 2.1}$ | $85.7_{\pm 1.5}$ | $78.6_{\pm 1.7}$ | $\underline{60.52}_{\pm 0.28}$ | $\underline{89.83}_{\pm 0.23}$ |
| *KPM* | *SMatchToPR* | *78.9* | *92.7* | *85.8* | – | – |
| *Baseline* | *NLP@UIT* | *74.6* | *93.0* | *83.8* | – | – |
| *(large)* | *Enigma* | *73.9* | *92.8* | *83.3* | – | – |

Table 1: Main result on KPM and CESC. FP: further pre-training; aug.: relative-augmented samples; mix: FP samples can either be plain or augmented, selected randomly during FP. The last model group shows KPM results from models with large-scale backbones. Bold font indicates the best results (except for the last group) under each evaluation metric; underscores indicate the best results within the model group (RoBERTa/GreaseArG). We also provide standard deviations of each metric across runs, shown next to the average.

from other models on this dataset, we use it to conduct experiments on KPM.

**CESC: IAM**  Besides the integrated task, Cheng et al. (2022) also proposed a benchmark dataset called IAM for training and evaluation. This dataset is generated from articles from 123 different topics. We adopt this dataset for CESC experiments.

### 5.3 Implementation

We conduct all experiments on NVIDIA GeForce RTX 4090 and repeat 4 times under different random seeds. In each run, the best model is chosen according to the performance on the dev set; the final result on the test set is the average of the ones from the best models in all 4 runs. Model and training configurations, including checkpointing rules, are listed in Appendix D.

To obtain further pre-training samples, especially augmented ones, we apply a few more operations on the pre-training dataset, details described in Appendix C. Note that documents from *args.me* naturally contain stance labels towards their conclusions. Therefore, we include the RSD task when pre-training with relative-augmented samples.

### 5.4 Baseline Models

**KPM**  A handful of participants in KPA-2021 have described their method for KPM, with which

we compare our models. These methods can be categorized into two groups: sentence pair classification predicts results on concatenated input, and contrastive learning uses the similarity between argument and key point representations.

**CESC**  Cheng et al. (2022) proposed and examined two approaches for CESC, pipeline and multi-label. The pipeline approach first extracts claims from articles and then decides their stances. The multi-label one treats this task as a 3-class classification task, where each candidate argument can be positive, negative, or unrelated to the given topic.

### 5.5 Main Results

Table 1 lists the main results on both downstream tasks. We use pair classification for our models in KPM experiments, which gives better results than contrastive learning. For KPM, we select 5 best results from Friedman et al. (2021): SMatchToPR (Alshomary et al., 2021) and MatchTstm (Phan et al., 2021) use contrastive learning; NLP@UIT, Enigma (Kapadnis et al., 2021) and ModernTalk (Reimer et al., 2021) use pair classification. For CESC, we select the results of both approaches mentioned in Cheng et al. (2022).

GreaseArG, after further pre-training with a portion of samples augmented, showed comparable (on KPM) or better (on CESC) performance selected base-scale baselines on either task. While

| Model | Support $F_1$ | Contest $F_1$ |
|---|---|---|
| RoBERTa | 40.43 | **47.74** |
| GreaseArG | **40.59** | 46.25 |

Table 2: $F_1$ scores on support/contest claim class on CESC. Here, we only show results from FP-with-mix models.

| Model | | Strict | Relaxed | Mean |
|---|---|---|---|---|
| MatchTstm | | 74.5 | 90.2 | 82.4 |
| RoBERTa | no FP | 67.3 | 83.9 | 75.6 |
| | FP w/o aug. | 69.7 | **86.4** | 78.0 |
| | FP w/ aug. | 69.1 | 85.1 | 77.1 |
| | FP w/ mix | **70.0** | 86.3 | **78.1** |
| GreaseArG | no FP | 63.7 | 81.1 | 72.4 |
| | FP w/o aug. | 65.6 | 81.7 | 73.7 |
| | FP w/ aug. | **69.6** | **86.6** | **78.1** |
| | FP w/ mix | 68.3 | 84.3 | 76.3 |

Table 3: Evaluation result on KPM, using contrastive learning instead of pair classification. Bold font indicates the best results within each group.

vanilla RoBERTa gives worse results[2] than baseline models, on CESC it gains a significant boost when further pre-trained with mixed samples. Our results being compared are the mean of 4 runs; when taking random fluctuations into account, further pre-trained GreaseArG can easily supersede base-scale models and even be on par with large-scale baselines.

# 6 Analysis

## 6.1 GreaseArG vs. Vanilla LM

As seen in Table 1, under the same further pre-training settings, GreaseArG significantly outperforms vanilla RoBERTa on all metrics of KPM. However, on CESC, the order is reversed. Table 2 demonstrates the $F_1$ scores of each stance class (support and contest). While GreaseArG supersedes RoBERTa concerning supporting claims, its performance on contesting claims is not as good as RoBERTa's. This result aligns with the ones on KPM, where GreaseArG shows an advantage since matching pairs of arguments and key points must have the same stance toward the topic.

[2]For CESC, RoBERTa without further pre-training is equivalent to the multi-label model. However, we use different random seeds and average results across all runs; hence, the final $F_1$ scores are different from what is reported by Cheng et al. (2022).

| Model | Macro $F_1$ | Micro $F_1$ |
|---|---|---|
| GreaseArG, FP w/ mix | 60.52 | 89.83 |
| w/o MNM | 59.87(−0.65) | 89.54(−0.29) |
| w/o MEM | 59.83(−0.69) | 89.78(−0.05) |
| w/o GCL | 60.36(−0.16) | 89.92(+0.09) |
| w/o TOP | 59.76(−0.76) | 89.91(+0.08) |
| w/o DIR | 59.67(−0.85) | 89.48(−0.35) |

Table 4: Pre-training task ablation results on CESC, based on GreaseArG, further pre-trained with mixed samples. Only graph-related tasks are considered.

| Model | Support $F_1$ | Contest $F_1$ |
|---|---|---|
| GreaseArG, FP w/ mix | 40.59 | 46.25 |
| w/o GCL | 40.04(−0.55) | 46.26(+0.01) |
| w/o TOP | 38.51(−2.08) | 45.96(−0.29) |

Table 5: $F_1$ scores on support/contest claim class on CESC when ablating GCL/TOP.

## 6.2 Relative-Augmented Further Pre-training

As shown in Table 1, adding samples augmented with relatives generally helps improve further pre-training. This phenomenon is more significant with GreaseArG and on the CESC task. Interestingly, models could prefer various portions of augmented samples for different downstream tasks: both vanilla RoBERTa and GreaseARG prefer mixed samples during further pre-training on CESC, yet this strategy has shown adverse effects on KPM.

## 6.3 Alternative Approach for KPM

We further investigate the impact of different downstream approaches. Table 3 illustrates the results on KPM under the same setup but using contrastive learning as the downstream approach. Since the model computes the representation of arguments and key points separately, it cannot exploit the potential graph connections between them, which could be why GreaseArG is worse than RoBERTa. Nonetheless, further pre-training benefits both models, where the gap between GreaseArG and RoBERTa nearly disappeared. Although the base-scale MatchTstm (Phan et al., 2021) from Friedman et al. (2021) show better results, this model has introduced other strategies to improve model performance and thus cannot be compared directly.

## 6.4 Contributions of Pre-training Tasks

We conduct ablation experiments on CESC using GreaseArG, further pre-trained with mixed

| Argument | Candidate Key Point | Gold Label | Hi-ArG Jaccard | RoBERTa, no FP | GreaseArG, FP w/ aug. |
|---|---|---|---|---|---|
| I do not agree to force children without parental consent should not be fair | Mandatory vaccination contradicts basic rights | non-matching | 0.0417 | **0.001** | 1.263 |
| | The parents and not the state should decide | matching | 0.1 | $-0.132$ | **1.609** |

Table 6: KPM sample where GreaseArG corrects vanilla RoBERTa. Hi-ArG Jaccard is the Jaccard Similarity between intra-arg node sets of argument and key point. Note that predicted scores across models are not comparable.

| Model | | KPM (mAP) | | | CESC ($F_1$) | |
|---|---|---|---|---|---|---|
| | | S. | R. | M. | Macro | Micro |
| RoBERTa | FP w/ mix | 72.2 | 87.6 | 79.9 | 61.07 | 90.28 |
| GreaseArG | FP w/ aug. | 75.8 | 89.5 | 82.6 | 59.65 | 89.61 |
| ChatGPT | Direct | 24.0 | 35.2 | 29.6 | 34.29 | 56.73 |
| | Explain | 43.5 | 58.8 | 51.2 | 41.60 | 69.02 |

Table 7: ChatGPT performance on KPM and CESC. Direct: the model generates the prediction directly. Explain: the model generates an explanation before the final prediction. Table headers are the same as Table 1.

samples, to measure the contributions of various graph-related pre-training tasks during further pre-training (the importance of MLM is obvious). Results are listed in table 4. All ablated models perform worse concerning the macro $F_1$. At the same time, a few of them have higher micro $F_1$, indicating that such ablation improves performance in some classes yet harms others. Tasks related to graph components contribute more; conversely, removing Graph Contrastive Learning (GCL) causes a less significant effect.

To further investigate the increasing micro $F_1$ when ablating GCL and Top Order Prediction (TOP), table 5 illustrates $F_1$ scores on claim classes when ablating either task. Ablation caused a significant drop in support $F_1$ yet did not significantly increase contest $F_1$. This indicates that the higher micro $F_1$ is due to the imbalance between claims and non-claims (527 against 6, 538): ablating these tasks might slightly improve *claim extraction* but could harm *stance classification*.

### 6.5 Hi-ArG Information Benefit

To examine the benefits of Hi-ArG information on downstream tasks, we compare model predictions on the KPM task, which is the candidate key point with the highest predicted score for each argument. We chose RoBERTa without further pre-training, and GreaseArG further pre-trained with augmented samples as target models. Among all 723 argu-

ments, 94 have seen the label of their predicted key point changed, of which 58 or 61.7% have positive changes. Table 6 demonstrates such a positive example. We can see that the argument and candidate key points have common Hi-ArG nodes, which GreaseArG can exploit.

### 6.6 Comparing with LLM

Large language models (LLMs) like ChatGPT and GPT-4 have recently shown competitive performance on various applications without fine-tuning (OpenAI, 2023). In response to this trend, we conduct primitive experiments that apply ChatGPT (gpt-3.5-turbo) on KPM and CESC, using in-context learning only. For KPM, we ask the model to give a matching score to maintain the evaluation process. For CESC, we asked the model to treat the task as a 3-class classification and predict the class label. All prompts consist of the task definition and several examples.

Table 7 demonstrates the results, comparing them with our best results. ChatGPT gives poor results when asked to predict directly, even with examples given. Asking the model to explain before answering can improve its performance, but it has yet to reach our models and other baselines. While these primitive experiments have not involved further prompt engineering and techniques, they have shown that more efforts (such as Hi-ArG) are needed for LLMs to handle computational argumentation tasks.

## 7 Conclusion

In this paper, we propose a new graph structure for argumentation knowledge, Hi-ArG. We design an automated construction pipeline to generate the lower intra-arg graph. To exploit information from Hi-ArG, we introduce a text-graph multi-modal model, GreaseArG, and a novel pre-training framework. On two different argumentation tasks (KPM and CESC), these approaches create models superseding current models under the same scale.

## Limitations

Despite the above results, this paper has a few limitations for which we appreciate future studies. First, the quality of Hi-ArG can be further improved using more powerful models and tools. Second, we only test on small language models due to resource limitation, yet, we conjecture that integrating Hi-ArG and LLMs could also be a potential solution to some current problems in LLMs, such as hallucination.

## Acknowledgments

This work is supported by National Natural Science Foundation of China (No. 6217020551) and Science and Technology Commission of Shanghai Municipality Grant (No.21QA1400600).

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

## A  AMR and Hi-ArG Intra-arg Graphs

In semantic analysis, Abstract Meaning Representation (AMR, Banarescu et al. 2013) is a highly abstract structure representing semantic information of sentences and documents. An AMR is a rooted, directed graph with labeled nodes and edges, where nodes record concepts or predicates and edges record semantic relations between them. In this way, sentences semantically identical can be assigned to the same AMR graph, even if they are syntactically different.

In a general AMR graph, concepts (as nodes) can be English words, PropBank (Palmer et al., 2005) framesets (a predicate labeled with semantic arguments), or predefined keywords such as negation (-); relations (as directed edges) can be frame arguments defined in PropBank, or other relations that predicate phrases cannot cover (domain, topic, quantity, date/time, list element, etc.).

For example, in Figure 5, both sentences can be represented by the given AMR graph. Concepts in the graph include gun, person, and kill-01, where kill is a PropBank frameset and kill-01 is one of its role sets, meaning "causing to die". Relations in the graph include :ARG0 and :ARG1, following the roleset definition of kill-01 where :ARG0 the killer and :ARG1 the corpse.

Due to its capability to express the semantics of multiple sentences compactly, we introduce AMR as the primary backbone of the intra-arg graph of Hi-ArG. More specifically, an argument is stored in the intra-arg graph in its AMR form, keeping all nodes and edges unchanged. Moreover, each AMR graph has a root node that can visit all of the nodes in the graph through the directed edges;

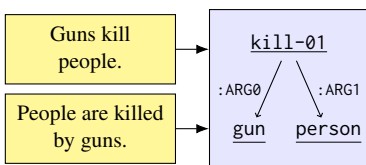

Figure 5: An example of an AMR graph representing two sentences: "Guns kill people" and "People are killed by guns."

thus, we choose this root node as the top node of the argument so that its descending subgraph is exactly the AMR graph of the argument. Finally, we force every AMR graph recorded in Hi-ArG to be acyclic, following the strict definition of AMR graphs, since every edge in an AMR graph can be reverted by appending to or removing from its relation label a passive mark -of.

## B  Hi-ArG Construction Details

### B.1  Extracting Stage

We use spaCy[3] to cut documents into sentences. Some basic rules can be applied to filter out invalid sentences for most corpus, such as limiting the minimum number of words or forcing all characters to be printable. While advanced filtering based on argumentation mining models can also be applied to enhance argument quality, in this paper, we only focus on basic filters, which are easier to implement.

### B.2  Parsing Stage

AMR parsing can be tedious when conducted manually; however, the recent development of language models has made automatic AMR parsing possible on large corpora. Although the AMR graphs generated by these models could contain noise, the negative effect is acceptable in specific applications such as pre-training (Bai et al., 2022). In this paper, we apply a BART-based (Lewis et al., 2020) model implemented by amrlib[4] to this stage to generate AMR graphs for all datasets.

### B.3  Merging Stage

We implement the iterative merging method in this stage based on node isomorphism. To identify isomorphic nodes, each node is labeled by its attribute (text), its child nodes, and the attributes of edges between them (referred to as child edges). Two nodes are called directly isomorphic if they share

---

[3] https://spacy.io/
[4] https://github.com/bjascob/amrlib

| # sent. | # token | # node | # edge | # top. |
|---------|---------|--------|--------|--------|
| 6.08M   | 141M    | 29.4M  | 61.0M  | 5.23M  |

Table 8: Statistics of the pre-training dataset constructed from the *args.me* corpus. The number of tokens is computed based on the RoBERTa tokenizer.

the same node attribute, child nodes, and child edge attributes — therefore must have identical labels. Note that merging nodes may only change the label of their entrance nodes, as AMR graphs are acyclic. Meanwhile, merging directly isomorphic nodes can make these nodes, if isomorphic, directly isomorphic. Thus, all isomorphic sub-graphs can be merged by repeatedly merging directly isomorphic nodes.

## C  Further Pre-training Details

### C.1  Hi-ArG Construction

In *args.me*, documents are recorded as conclusion–premise pairs, where the premises can be for or against the conclusion. During the extracting stage, premise sentences shorter than 5 words or contain unprintable characters are excluded; the conclusion sentence, leading the whole document, is rewritten as *"..." is right/wrong.*[5] The parsing and merging stages are the same as described in Section 3.2, yet we also include an extra graph-text aligning using `amrlib` for multi-modal mask generation.

The final further pre-training dataset contains $6,080,249$ sentences and a large Hi-ArG. More detailed statistics of the dataset are listed in Table 8. Note that several sentences share the same meaning. Hence each group of such sentences corresponds to one single top node.

### C.2  Relative Searching

As mentioned in Section 4.3, we can search for relatives in Hi-ArG with the help of the sentence-node graph. Since the full sentence-node graph of *args.me* is too large to perform a weighted sampling for all sentences, we introduce a few constraints on candidate sentence pairs to reduce computation:

1. Sentences must not share one top node in the original Hi-ArG.

2. Sentences must come from documents under the same topic or conclusion.

3. Sentences must not appear too close in the source corpus: there should be at least $L$ sentences between them.

4. Only nodes connected to no more than $S$ sentences are considered.

5. Leading hint sentences (*"..." is right/wrong.*) will not have sampled relatives, though they can be relatives of other sentences.

The third constraint is to prevent relatives from appearing in the training sample twice, and the fourth is to exclude pairs with only weak connections through "public" nodes representing common concepts. In this paper, we use $L = 31$ and $S = 500$. Although several sentences cannot find any relative under these constraints, $32.10\%$ of all sentences have at least one candidate match.

Due to the limitation of computational resources, we sample relatives before pre-training, among which $45.88\%$ are of the same stance. This ratio is balanced enough for a valid RSD task for further pre-training.

### C.3  Multi-modal Mask Generation

Although random masking can help language models learn text representations, improving mask selection with extra information from Hi-ArG is possible. Here we propose a special masking procedure to generate more valuable masks on both texts and graphs (nodes and edges):

1. Masks are generated on graphs, where the probability of each node being masked is proportional to the number of sentences it is related to, and of each edge, the weight product of nodes on both sides. In this way, common nodes and edges across sentences are more likely to be masked and let the model focus on such intersections. The overall mask ratio can still be controlled at a specific level.

2. A pre-mask on text is computed based on graph masks, achieved using graph-text alignments generated beforehand. Each masked node and each node with a masked edge looks for any aligned token spans and mask them, creating a base text mask that shall not leak information between text and graph.

3. If the pre-mask does not reach the desired mask ratio for text, extra masks will be generated on non-mask tokens to satisfy the requirement.

---

[5]For instance, if a document supports the conclusion *We should ban guns*, then the first sentence extracted from the document will be *"We should ban guns" is right.*

| Model Configuration | Value |
|---|---|
| Maximum # Tokens | 512 |
| Node PE Dimension | 4 |
| # Joint Blocks | 9 |
| GNN Hidden Size | 256 |
| Graph Transformer Attention Heads | 2 |
| # Mix Layers | 1 |
| Mix Layer Hidden Size | 512 |
| Cross-modal Attention Heads | 8 |

Table 9: Model configuration of GreaseArG. Configurations not mentioned here inherit values from RoBERTa-base and Graph Transformer.

| Pre-training Configuration | Value |
|---|---|
| Batch Size | 32 |
| # Tokens per Sample | 512 |
| Optimizer | AdamW |
| LM Learning Rate | $5 \times 10^{-5}$ |
| Non-LM Learning Rate | $1 \times 10^{-4}$ |
| LR Decay | inverse_sqrt |
| Warmup Steps | $2\,500$ |
| Total Steps | $180,000$ |

Table 10: Pre-training configuration for both RoBERTa and GreaseArG.

# D  Implementation Details

## D.1  Model

Table 9 lists model configuration parameters applied to all experiments. Graph Transformer requires positional embeddings (PE) for every node to encode positional information. Following the suggestions by Dwivedi and Bresson (2020), laplacian eigenvectors are chosen as node PEs, calculated on a bi-directed graph removing all edge attributes.

## D.2  Further Pre-training

Table 10 lists hyper-parameters used during further pre-training, applied to both RoBERTa and GreaseArG. Training samples are guaranteed to fill up 512 tokens by concatenating as many sentences as possible. The total training step is approximately one epoch concerning the training set ($95\%$ of all pre-training data), rounding to ten thousand for checkpoint generation.

## D.3  Fine-tuning

Fine-tuning hyper-parameters for the two downstream tasks are listed in Table 11.

## D.4  Checkpointing Rules

For each pre-training model and configuration, checkpoints are saved every $10\,000$ steps, and these

| Fine-tuning Configuration | KPM | CESC |
|---|---|---|
| Batch Size | 32 | 128 |
| Optimizer | AdamW | AdamW |
| Learning Rate | $2 \times 10^{-5}$ | $4 \times 10^{-5}$ |
| LR Decay | inverse_sqrt | inverse_sqrt |
| Warmup Ratio | 0.1 | 0.1 |
| Total Epochs | 20 | 20 |

Table 11: Fine-tuning configuration for KPM and CESC.

checkpoints will be used as base models during fine-tuning. In the fine-tuning stage, after each fine-tuning epoch, evaluation will be conducted on dev and test sets. Test results from the model with the highest gold metric on the dev set will be chosen as the final test result of the current random seed. The mean of strict and relaxed mAP is selected for KPM as the gold metric; for CESC, the macro $F_1$ is selected.