# OpenReview forum: "Hi-ArG: Exploring the Integration of Hierarchical Argumentation Graphs in Language Pretraining"
_EMNLP/2023/Conference — EMNLP 2023 Main_

### Official Review · Reviewer_TQxT · 2023-07-28

**Typos Grammar Style And Presentation Improvements:** n/a
**Soundness:** 3

**Excitement:**

4: Strong: This paper deepens the understanding of some phenomenon or lowers the barriers to an existing research direction.

**Missing References:**

n/a

**Paper Topic And Main Contributions:**

Problem: This paper deals with the problem of incorporating knowledge graph in argumentation tasks, including Key Point Matching and Claim Extraction with Stance Classification.

Contribution:
1. It proposes Hi-ArG, a graph structure based on Abstract Meaning Representation (AMR) which is better than well-known Argumentation Knowledge Graph to retain both intra-level and inter-level info of arguments in documents and the whole corpus.
2. It exploits the new structure in two ways, model architecture and data augmentation. In term of model architecture, they modify GreaseLM to form GreaseArG model which facilitates the use of Hi-ArG in the GNN component of the model, as long as propose a new set of further-pretraining graph tasks for GreaseArG. In term of data augmentation, they utilize graph structure of Hi-ArG to find text relevant to input documents (called "relatives") which facilitates another pre-training objective for GreaseArG.
3. It conducts experiments, including constructing Hi-ArG graphs for further-pretraining/evaluation and training GreaseArG. The results show the improvement over same-scale models on the considered evaluation tasks.

**Questions For The Authors:**

1. What does "sampled documents" in line 343 refer to? I assume that in each training instance, there is only one document.

2. What is the effect of link node? I can think of an alternative w/o link node to solve two issues mentioned in Section 3.3: for the "disconnected" case, we just connect top nodes to the Root node, and for the "same top node" case, we replicate the top nodes and intermediate nodes in sub-graph if needed, then connect top nodes to the Root node.

**Reasons To Accept:**

This paper introduces a brand-new way to represent text as knowledge graphs and use it in argumentation tasks, which opens a highly potential direction to apply it in various tasks, including generation tasks such as question answering and knowledge-grounded generation.

The data structure (general or adapted) and the construction of corresponding Hi-ArG knowledge graph is clearly described. The experimental setup, including the new model architecture, pretraining tasks, and data augmentation method is also clear.

Although the improvement brought by Hi-ArG is not large and in fact it doesn't beat the SOTA task models, the improvement is significant over the same-scale models. Furthermore, the paper does extensively analyze the effect of each proposed components in the further-pretraining stage, help readers understand behaviors of the new model in evaluation tasks and the contribution of each proposed component.

**Reasons To Reject:**

The building block AMR is not re-introduced in the paper with important notes of authors, making the paper hard to follow continuously and hard to link AMR to the concepts in section 3.1 and 3.3, e.g top nodes.
Also, although the description of method is clear, because text is inherently not the best way to describe a structure, the lack of reference between Section 3.3 and Figure 3 makes it harder to understand how to add the link nodes.
Nonetheless, the reason "multiple sentences in one text can point to the same top node, which could confuse each other" to introduce link node is not clearly justified.

The paper does introduce the inter-arg level of Hi-ArG, but never construct Hi-ArG with it or use it in pretraining/inference. That confuses readers about the necessity of the inter-arg level of Hi-ArG, at least in the scope of this paper.

Since the paper focus on Hi-ArG, readers expect to see the comparison between Hi-ArG and AKG. However, the paper does not show the experiment result of GreaseLM - which is the AKG counterpart of GreaseArG - on the two evaluation tasks. Thus, it's unclear about the superiority of Hi-ArG over AKG.

**Reproducibility:**

3: Could reproduce the results with some difficulty. The settings of parameters are underspecified or subjectively determined; the training/evaluation data are not widely available.

**Reviewer Confidence:**

4: Quite sure. I tried to check the important points carefully. It's unlikely, though conceivable, that I missed something that should affect my ratings.

---

> ### Author Rebuttal · Authors · 2023-08-28
>
> We thank you for taking the time to review our submission and for your valuable feedback.
>
> In response to your comments and questions, we would like to provide the following rebuttal:
>
> ## Q1: Re-introduction of AMR and related concepts
>
> We appreciate your advice on a clear introduction to AMR concepts. In our paper, we followed the original definitions of AMR graphs so that we could apply existing parsers to automated Hi-ArG construction; yet, in response to your comment, we will include essential information (text and figures if necessary) around AMR and related concepts in the revised version of our paper.
>
> ## Q2: Clarity of section 3.3 and advantages of link nodes
>
> We thank you for pointing out the potential confusion in section 3.3 and will add the necessary explanations for this section. Yet, we would like to emphasize the advantages of link nodes introduced in our paper.
>
> As explained, link nodes are necessary to distinguish multiple appearances of a single argument (possibly with nuances) in the given text. While Hi-ArG does not store transitional information as the graph is context-independent, such contextual information could become important when cooperating with text data. In this case, link nodes can handle this information while leaving context-independent information (e.g., semantics and logic) to Hi-ArG nodes.
>
> Another advantage of link nodes is efficiency. Link nodes could reduce the number of new nodes and edges, especially when the sub-graph referred to multiple times contains a lot of nodes and edges. They also allow the sharing of information through one common sub-graph.
>
> Finally, both using link nodes and copying nodes need an extra root node to ensure graph connectivity, so we suggest no difference in this part.
>
> ## Q3: Necessity of inter-arg graphs
>
> We understand your concern about the inter-arg level of Hi-ArG.
>
> Due to the nature of the two downstream tasks we selected, we did not include inter-arg graphs in our downstream experiments to avoid leaking information. However, we would like to propose two potential applications of inter-arg graphs not mentioned in this paper.
>
> First, inter-arg graphs can also participate in the pre-training stage for more general uses so that pre-trained models can learn logical relations between arguments. Furthermore, when applying our proposed pre-training method with relatives, we can sample the relatives of a given argument from both intra-arg and inter-arg levels of Hi-ArG as long as other constraints are satisfied.
>
> Second, inter-arg graphs can provide clues for generative tasks like argument generation. These tasks require claims and evidence from multiple aspects, as well as multi-step induction/deduction among arguments. The Inter-arg graph is a proper source to retrieve such arguments and logical relations.
>
> ## Q4: Comparison between Hi-ArG and AKG
>
> We understand your comment concerning the comparison between Hi-ArG and AKG.
>
> Before discussing this topic, we would like to clarify that GreaseLM is not the AKG counterpart of GreaseArG: GreaseLM is a model designed for questioning answering tasks, whose authors used ConceptNet and a self-constructed graph for their experiment but not AKG; in our paper, we modified the structure of GreaseLM for pre-training purposes, resulting in GreaseArG, which outputs text/node representations for general use.
>
> For the main issue, because Hi-ArG and AKG have completely different structures, it would be better to compare these two graphs through downstream applications. However, we understand that AKG is currently not applied to the two downstream tasks we introduced (we only found applications on argument generation). Therefore, it requires novel methods to exploit AKG for certain tasks where Hi-ArG can also be applied to make a meaningful comparison possible. We would like to include this subject in future studies concerning more downstream applications of Hi-ArG.
>
> ## Q5: Definition of "sampled documents"
>
> As mentioned in the same section, a document consists of multiple sentences. When appearing in one training instance, these sentences form a continuous segment as we follow the sample construction method of RoBERTa.
>
> In our experiments, a document is usually a debate speech or something similar, and most of the documents can fit in a single training instance. Therefore, when we mention "sampled documents" or other similar terms, we refer to the documents within the training instance, complete or split at the beginning/end.

---

### Official Review · Reviewer_nF6N · 2023-08-04

**Soundness:** 4

**Excitement:**

4: Strong: This paper deepens the understanding of some phenomenon or lowers the barriers to an existing research direction.

**Paper Topic And Main Contributions:**

This paper proposes a new hierarchical structure Hi-ArG to represent argumentations. It can retain more semantics within arguments at the intra-argument level and record relations between arguments at the inter-argument level.

To exploit Hi-ArG, they propose several pretraining tasks to augment a text-graph model with the information from the argumentation graphs. Experiments on two argumentation tasks, Key Point Matching and Claim Extraction with Stance Classification, have shown that further pre-training and fine-tuning improve performance.

**Reasons To Accept:**

- A new structure to represent and store argumentations, which can facilitate future research on argumentation tasks.

- Two potential methods to use the structural information from Hi-ArG

- Intensive experiment to show the effectiveness of the proposed methods

**Reasons To Reject:**

- The performance boost brought by the proposed method is not very significant,

- Besides, I do not see major weaknesses that prevent the paper from being accepted. Yet this paper can be improved by providing the following results:

  - evaluation on the correctness of machine-generated Hi-ArG
  - the performance of LLMs such GPT3.5 (possibly combined with in-context learning) on the two evaluation tasks

**Reproducibility:**

3: Could reproduce the results with some difficulty. The settings of parameters are underspecified or subjectively determined; the training/evaluation data are not widely available.

**Reviewer Confidence:**

3: Pretty sure, but there's a chance I missed something. Although I have a good feel for this area in general, I did not carefully check the paper's details, e.g., the math, experimental design, or novelty.

---

> ### Author Rebuttal · Authors · 2023-08-28
>
> We thank you for taking the time to review our submission and for your valuable feedback.
>
> In response to your comments and questions, we would like to provide the following rebuttal:
>
> ## Q1: Insignificant performance boost
>
> We understand your concern regarding the significance of the performance boost. Yet we would like to provide a potential explanation on this point.
>
> We repeated all experiments four times with different seeds and averaged their results as the final ones demonstrated in our paper; however, the baseline results to which our results are compared may follow a different setting. For instance, in CESC, our RoBERTa, without further pre-training, achieved a lower average performance than the multi-label baseline, even though both models are *essentially the same*.
>
> ## Q2: Evaluation of automatically generated Hi-ArG
>
> We appreciate your advice regarding the correctness of the generated Hi-ArG.
>
> The quality of generated intra-arg and inter-arg graphs can be measured using the correctness of their corresponding sub-tasks (e.g., AMR parsing and argument mining). We believe current methods are powerful and robust enough to ensure *in-level* quality.
>
> However, as we understand, evaluating the *entire* Hi-ArG requires novel approaches since no existing metric exists. A potential way we would like to propose here is to use a small fraction of the corpus and measure the similarity between manually annotated and automatically generated graphs.
>
> The above method is only one of many possible approaches, and we encourage more research on this topic. Nevertheless, the evaluation task would be crucial in future work to improve Hi-ArG quality.
>
> ## Q3: LLM performance on downstream tasks
>
> We appreciate your advice concerning LLM's performance on our downstream tasks.
>
> We have conducted a few primitive experiments that apply `gpt-3.5-turbo` (GPT) on KPM and CESC, using in-context learning only. For KPM, we asked GPT to give a matching score so that the evaluation process remained the same. For CESC, we asked GPT to treat the task as a 3-class classification and predict the class label.
>
> We observed that GPT gave poor results when asked to predict directly, even with examples given. Asking GPT to explain before answering improved its performance, but it has yet to reach our models and other baselines.
>
> | Model                | (KPM) Strict mAP | Relaxed mAP | Mean of mAP | (CESC) Macro $F_1$ | Micro $F_1$ |
> | -------------------- | ---------------- | ----------- | ----------- | ------------------ | ----------- |
> | RoBERTa FP w/ mix    | $72.2$           | $87.6$      | $79.9$      | $61.07$            | $90.28$     |
> | GreaseArG FP w/ aug. | $75.8$           | $89.5$      | $82.6$      | $59.65$            | $89.61$     |
> | GPT (direct predict) | $24.0$           | $35.2$      | $29.6$      | $34.29$            | $56.73$     |
> | GPT (explain first)  | $43.5$           | $58.8$      | $51.2$      | $41.60$            | $69.02$     |
>
> While these primitive experiments have not involved further prompt engineering and techniques like chain of thoughts, they have shown that more efforts are needed for LLM to handle computational argumentation tasks like KPM and CESC. We hope that Hi-ArG can become one of these efforts in the future.

---

### Official Review · Reviewer_zpwA · 2023-08-05

**Paper Topic And Main Contributions:** 1) This paper introduces a Hierarchic…
**Typos Grammar Style And Presentation Improvements:** 1) line 966, comma is missing in numb…
**Soundness:** 5

**Excitement:**

4: Strong: This paper deepens the understanding of some phenomenon or lowers the barriers to an existing research direction.

**Missing References:**

[1] SpaCy in Appendix A.1

**Questions For The Authors:**

A. Why did you bold the performance excluding large-scale models' performance in Table 2? I think it's really confusing to reading the table.
B. Why is the performance drops when ablating GCL, TOP pre-training tasks in Table 6?

**Reasons To Accept:**

1) This work suggests Hi-ArG, and its graph structure is sufficient to retain more semantics within the arguments by borrowing AMR's structure and concepts.
2) Also, the exploitation methods of the Hi-ArG are well explained, and the experiments and analyses show its potential.
3) The paper also analyzes the degradation of the GreaseArG in detail in the analysis section which lead the way of further research.

**Reasons To Reject:**

1) The paper needs more clarity in notation. Especially, the explanation of the notations in Figure 3 is missing such as t^tilde, G, E, and V. Even the notations are widely used, please indicate what they are in the paper.
2) Visualization of graph structure tasks for the pre-training would be helpful. The three pre-training tasks need more explanation along with visualization for clear understanding.

**Reproducibility:**

4: Could mostly reproduce the results, but there may be some variation because of sample variance or minor variations in their interpretation of the protocol or method.

**Reviewer Confidence:**

4: Quite sure. I tried to check the important points carefully. It's unlikely, though conceivable, that I missed something that should affect my ratings.

---

> ### Author Rebuttal · Authors · 2023-08-28
>
> We thank you for taking the time to review our submission and for your valuable feedback.
>
> In response to your comments and questions, we would like to provide the following rebuttal:
>
> ## Q1: Clarity in notations, especially in Figure 3
>
> We thank you for pointing out the need for more clarity in the notation. In Figure 3, $\tilde{t}$ denotes token embeddings went through at least one LM layer, distinguishing from the original $t$ generated by the token embedding layer. $G=\langle\tilde{V},E\rangle$ denotes the embedded graph went through at least one GNN layer: $\tilde{V}=\{\tilde{v_1},\tilde{v}_2,\tilde{v}_3,\dots,\}$ are node embeddings transformed by at least one GNN layer, also distinguishing from the original $V=\{v_1,v_2,v_3,\dots\}$ generated by the node embedding layer; $E=\{e_0,e'_0,e_1,e'_1,\dots\}$ are edge embeddings generated by the edge embedding layer (prime marks indicate reversed edges), which remain unchanged throughout the process.
>
> In response to your comment, we will add further explanations of several notations that may confuse, including the ones you have mentioned, to the revised version of our paper.
>
> ## Q2: Further explanation with visualization for pre-training tasks
>
> We appreciate your advice concerning visualization. We will add more details about pre-training tasks to the Appendix in the revision, including necessary text and figures.
>
> ## Q3: Notations and format of Table 2
>
> We mainly compared our model with base-scale baseline models because they have the same backbone capability and pre-training knowledge. Meanwhile, the KPM-2021 shared task's ranking contains both base-scale and large-scale models, and we found that our model performs on par with some large-scale models. Therefore, we decided to include these baseline results in Table 2.
>
> Yet, in response to your concern, we will split the KPM baselines into base-scale and large-scale groups in the revised version of Table 2 to further clarify it.
>
> ## Q4: Model performance when ablating GCL or TOP
>
> We suppose you are concerned about the increasing micro $F_1$ when ablating GCL and TOP. However, the increments are statistically insignificant, and their macro $F_1$ decreases more (especially for TOP). We suggest this is due to the imbalance between claims and non-claims (much more than claims): ablating these tasks might *slightly* improve **claim extraction** but could harm **stance classification**.
>
> Further investigations have shown that when considering claim stance classes, ablating either task caused a significant drop in support $F_1$, yet did not significantly increase contest $F_1$. See the table below for evaluation details. These results support our claim above.
>
> | Model                | Macro $F_1$       | Micro $F_1$       | Support $F_1$     | Contest $F_1$     |
> | -------------------- | ----------------- | ----------------- | ----------------- | ----------------- |
> | GreaseArG, FP w/ mix | $60.52$           | $89.83$           | $40.59$           | $46.25$           |
> | ... w/o GCL          | $60.36$ ($-0.16$) | $89.92$ ($+0.09$) | $40.04$ ($-0.55$) | $46.26$ ($+0.01$) |
> | ... w/o GCL          | $59.76$ ($-0.76$) | $89.91$ ($+0.08$) | $38.51$ ($-2.08$) | $45.96$ ($-0.29$) |
>
> ## Q5: Missing reference, typo, and misplaced Table 1
>
> We thank you for notifying us about these referencing and writing issues. We will fix them in the revised version of our paper.

---

### Meta-Review · Area_Chair_t13Q · 2023-09-26

**Recommendation:** 4

**Metareview:**

This paper proposes a new hierarchical structure Hi-ArG to represent argumentations. It can retain more semantics within arguments at the intra-argument level and record relations between arguments at the inter-argument level. To exploit Hi-ArG, they propose several pre-training tasks to augment a text-graph model with the information from the argumentation graphs. Experiments on two argumentation tasks, Key Point Matching and Claim Extraction with Stance Classification, have shown that further pre-training and fine-tuning improve performance.

In general, the authors found the proposed Hi-ArG to be an interesting method, the claim that it retains more semantics within the arguments at the intra-/inter-argument level is well-supported and the intensive experiment shows the effectiveness of proposed methods and the effects of each proposed component.

The more significant reasons to reject seem to be related to clarification and requests for additional results.
* Requests for clarification: the rebuttal provided a decent expansion on the issues brought up by reviewers.
* Requests for additional results: the authors provided additional results for one request (results on LLMs on the two evaluation tasks), however other requests were deferred to future work.

---

### Decision · Program_Chairs · 2023-10-07

**Decision:**

Accept-Main

**Comment:**

This paper proposes a new hierarchical structure Hi-ArG to represent argumentations. It can retain more semantics within arguments at the intra-argument level and record relations between arguments at the inter-argument level. To exploit Hi-ArG, they propose several pre-training tasks to augment a text-graph model with the information from the argumentation graphs. Experiments on two argumentation tasks, Key Point Matching and Claim Extraction with Stance Classification, have shown that further pre-training and fine-tuning improve performance.

In general, the authors found the proposed Hi-ArG to be an interesting method, the claim that it retains more semantics within the arguments at the intra-/inter-argument level is well-supported and the intensive experiment shows the effectiveness of proposed methods and the effects of each proposed component.

The more significant reasons to reject seem to be related to clarification and requests for additional results.
* Requests for clarification: the rebuttal provided a decent expansion on the issues brought up by reviewers.
* Requests for additional results: the authors provided additional results for one request (results on LLMs on the two evaluation tasks), however other requests were deferred to future work.